

# A Simulation Approach to Characterizing Sub-Glacial Hydrology

Chris Pierce[1], Christopher Gerekos[2], Mark Skidmore[3], Lucas Beem[3], Don Blankenship[2], Won Sang Lee[4], Ed Adams[1], Choon-Ki Lee[4], and Jamey Stutz[2]

1. Department of Civil Engineering, Montana State University, Bozeman Montana, USA

2. Institute for Geophysics, University of Texas at Austin, Austin Texas, USA

3. Department of Earth Sciences, Montana State University, Bozeman Montana, USA

4. Division of Glacial Environment Research, Korea Polar Research Institute, Seoul, South Korea

**Correspondence:** Chris Pierce (christopherpierce3@montana.edu)

**Abstract.** The structure and distribution of sub-glacial water directly influences Antarctic ice mass loss by reducing basal shear stress and enhancing grounding line retreat. A common technique for detecting sub-glacial water involves analyzing the spatial variation in reflectivity from an airborne ice penetrating radar (IPR) survey. Basic IPR analysis exploits the high dielectric contrast between water and most other substrate materials, where a reflectivity increase $\geq 15\,dB$ is frequently correlated with

the presence of sub-glacial water. There are surprisingly few additional tools to further characterize the size, shape, or extent of hydrological systems beneath large ice masses.

We adapted an existing radar backscattering simulator to model IPR reflections from sub-glacial water structures using the University of Texas Institute for Geophysics (UTIG) Multifrequency Airborne Radar Sounder with Full-phase Assessment (MARFA) instrument. Our series of hypothetical simulations modeled water structures from $5m$ to $50m$ wide, surrounded by

bed materials of varying roughness. We compared the relative reflectivity from rounded Röthlisberger channels and specular flat canals, showing both types of channels exhibit a positive correlation between size and reflectivity. Large ($> 20m$), flat canals can increase reflectivity by more than $20\,dB$, while equivalent Röthlisberger channels show only modest reflectivity gains of $8 - 13\,dB$. Changes in substrate roughness may also alter observed reflectivity by $3 - 6\,dB$. All of these results indicate that a sophisticated approach to IPR interpretation can be useful in constraining the size and shape of sub-glacial water, however a

highly nuanced treatment of the geometric context is necessary.

Finally, we compared simulated outputs to actual reflectivity from a single IPR flight line collected over Thwaites Glacier in 2022. The flight line crosses a previously proposed Röthlisberger channel route, with an obvious bright bed reflection in the radargram. Through multiple simulations, we demonstrated the important role that topography and water geometry can play in observed IPR reflectivity. We ultimately conclude the bright reflector from our IPR flight line is more likely a broad area

of wide distributed water, such as a series of flat canals or sub-glacial lake, instead of a Röthlisberger channel. The approach outlined here has broad applicability for studying the basal environment of large glaciers, and can aid in constraining the geometry and extent of sub-glacial hydrologic structures.



## 1 Introduction

The size, shape, and distribution of sub-glacial water is important to ice dynamics, and remains a significant uncertainty in
projecting sea level rise due to ice mass loss. Hydrological structures directly influence basal shear stress distribution, which
defines the boundary condition for rheology at the ice/bed interface (Schroeder et al., 2013; Tulaczyk et al., 2000; Church et al.,
2013). Widely distributed water has been shown to lubricate the base and weaken sediments (Dunse et al., 2015; Schoof, 2010).
Conversely, narrow water channels may have little impact on shear stress at the bed (Schroeder et al., 2013), but act as conduits
for concentrating meltwater produced upglacier. The size and location of such channels can directly influence grounding line
retreat by controlling the volume of water transported to the grounding line (Young et al., 2016; Wright et al., 2012; Schroeder
et al., 2013).

Airborne Ice Penetrating Radar (IPR) is an established technique for studying sub-glacial hydrology throughout Earth's
cryosphere. Attenuation loss through ice at common radar wavelengths ($\sim 2-5\,m$) is relatively low, enabling reliable imaging
of bed surfaces beneath ice masses several thousand meters thick. The dielectric contrast at an ice / water interface is much
higher than at an ice / rock interface, resulting in higher bed reflectivity when water is present. Additionally, some ice /
water interfaces may be relatively smooth, depending on the geometry of the hydrological structure. These properties are
commonly exploited in IPR surveys to infer the location of sub-glacial water, based on reflected power $> \sim 15\,dB$ higher than
the surrounding area (Schroeder et al., 2015; Peters et al., 2005; Rutishauser et al., 2018; Schroeder et al., 2013; Young et al.,
2016).

Multiple hypotheses pertaining to the sub-glacial environment beneath Thwaites Glacier in West Antarctica are derived from
IPR surveys. For example, Schroeder et al. (2013) first postulated a hydrological system dominated by channelized drainage
within $\sim 45\,km$ of the Thwaites grounding line. This hypothesis is based on data interpretation from the extensive 2003-2004
Airborne Geophysics of the Amundsen Sea Embayment, Antarctica (AGASEA) radar survey. Hager et al. (2022) ran a suite
of sub-glacial hydrology simulations to evaluate the probability of persistent channelization routes beneath Thwaites. Their
analysis concluded that Thwaites' near terminus hydrology is most likely comprised of a few persistent, high volume channels
flowing toward the central grounding zone. Two probable routes were proposed, as shown in Fig. 1.

Deducing the presence of water beneath a glacier such as Thwaites using IPR is relatively common, however methods for
testing further hypotheses regarding size, geometry, or hydrological structure remain challenging. Direct observation of the
glacier bed over any significant spatial extent is infeasible with current methods (e.g. drilling (Priscu et al., 2021)), limiting our
ability to calibrate radar returns to observed hydrological features.

This problem has precedent in inter-planetary science, where radar experiments are designed to test hypotheses with lim-
ited in situ evidence about surface or sub-surface characteristics. Backscattering simulators have proven especially useful in
modeling radar returns for celestial targets (Spagnuolo et al., 2011; Russo et al., 2008; Gerekos et al., 2018). Most simulators
approximate a target surface as a series of flat facets acting as point backscatterers. The backscattered electric field strength at
the radar antenna position is estimated with common mathematical approximations such as the Stratton-Chu integral. In such
a point-scatterer formulation, facets much smaller than the radar wavelength (typically $< \lambda/10$) are required to approximate





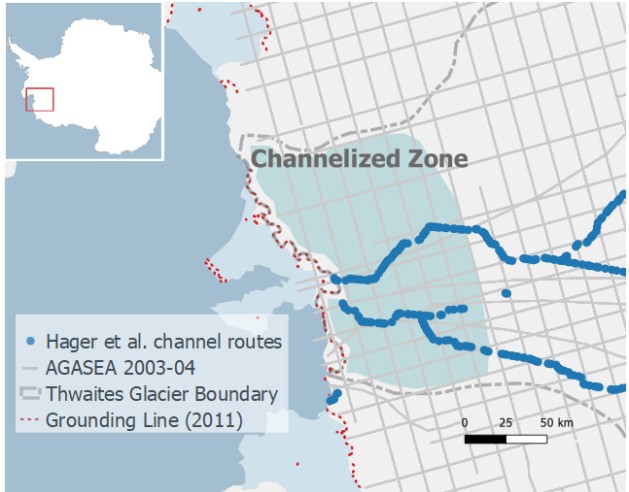

**Figure 1.** The grounding line region of Thwaites Glacier, with region of channelized hydrology proposed by Schroeder et al. (2013) high-lighted. AGASEA radar survey lines are shown in light grey. Likely high volume channel routes suggested by Hager et al. (2022) are shown in blue.

real world instrument results (Gerekos et al., 2018). This constraint necessitates access to high-end computing resources, often making point-scattering radar simulators unrealistic for IPR modeling.

However, as algorithms improve and computational costs decrease, it is increasingly attractive to attempt such a simulation
method with IPR radar / target problems. Gerekos et al. (2018) described a simulation technique that is particularly intriguing for the study of sub-glacial hydrology. The methodology is unique in two distinct ways from other simulators, which are very helpful in modeling IPR. First, the simulator can estimate strength and direction of signals transmitted through multiple layered material interfaces. This makes it conducive to targets such as the ice / bedrock system. Second, the algorithm allows phase to vary linearly across the facet (termed the Linear Phase Approximation or LPA). This feature enables modeling with
significantly larger facets ($\sim \lambda$ or larger), drastically reducing computational resources needed for accurate simulations.

This paper demonstrates the application of this radar simulation methodology to geometric scattering by common hydrological targets: flat canals and Röthlisberger channels. We discuss the relevant parameters to achieve accurate model results, and illustrate its utility in interpreting radar signatures from hydrological features beneath the ice. Finally, we apply the simulator to a hydrological target beneath Thwaites Glacier to constrain its geometry and extent.

## 2 Methodology

### 2.1 Simulation Setup

Figure 2 describes a generic conceptual model for an IPR radar simulation. Parameters were chosen to emulate a typical helicopter-based airborne radar survey with the University of Texas Institute for Geophysics (UTIG) Multifrequency Airborne



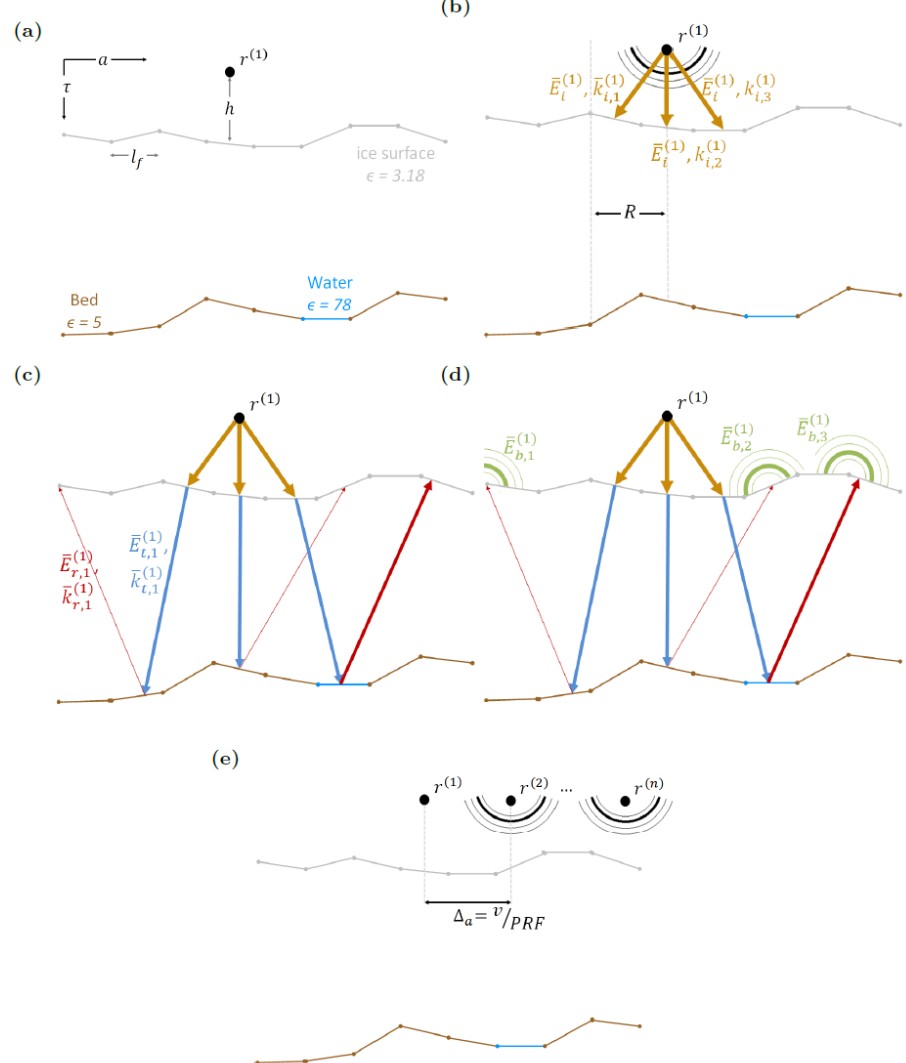

**Figure 2.** 2-D Conceptual model of an IPR radar simulation, defined in the direction of flight: a) The material and geometric model for a radar observation. b) The radar's spherical wavefront is approximated as rays directed at facets within a footprint beneath the aircraft. The simulator calculates a reflected and transmitted field strength, based on the dielectric constant at the surface ($\epsilon_{ice}$). We do not show the surface reflection for simplicity. c) The rays propagate through the ice, with a direction obeying Snell's Law. At the bed, reflected field strength and direction are controlled by dielectric contrast and incidence angle with the basal facet. d) The backscattered electric field strength ($E_b$) is approximated with the Stratton-Chu integral. e) The aircraft's position is incremented and the process is repeated.

Radar Sounder with Full-phase Assessment (MARFA) instrument (Castelletti et al., 2017; Lindzey et al., 2020). An ice surface

and a bed surface are defined in a 3-dimensional Cartesian coordinate system, noted as $S_{ice}$ and $S_{bed}$ respectively. Figure 2a illustrates a 2-D representation of these surfaces, divided into $N$ facets using a Delaunay triangulation algorithm, with





characteristic length $l_f$. The linear phase approximation (LPA) employed in Gerekos et al. (2018) allows the phase of the incident and reflected electric fields to vary linearly across each facet. LPA enables accurate simulations of coherent radar using relatively large facets ($\sim \lambda$ vs. $\lambda/10$ with other methods). The facet size should be constrained according to Eq. 1, where $h$ is the aircraft height and $\lambda$ is the free space radar wavelength (Gerekos et al., 2018). For simulations presented here, $h = 500m$ and $\lambda = 5m$, consistent with a typical UTIG MARFA helicopter experiment. $l_f = 5m$ was chosen, consistent with Eq. 1.

$$l_f \leq 0.2\sqrt{\lambda h/2} \tag{1}$$

The radar's spherical wavefront is simulated as a series of plane waves, with wavevectors $\bar{k}_i$ and field strength vector $\bar{E}_i$ directed at each ice surface facet within a radius $R$ beneath the aircraft (Fig. 2b). A wider $R$ will provide a more complete approximation of the radiated and returned electric field, but comes at a computational cost proportional to $R^2$. Therefore, $R$ is chosen to balance these competing priorities, as we will discuss in detail.

Each ray's path is traced through transmission at $S_{ice}$ using Snell's Law, then reflection from $S_{bed}$ (Fig. 2c). In Fig. 2, the reflection from the ice surface is omitted for simplicity, since we are most concerned with reflections from the bed in this study. Reflected and transmitted field strengths ($\bar{E}_t$ and $\bar{E}_r$) are calculated from the real component of the dielectric constant at $S_{ice}$ and $S_{bed}$, as discussed below. Attenuation loss between the surfaces is dependent on the complex component of the dielectric constant for ice (Gerekos et al., 2020), which is assumed constant over the short simulation distances presented here.

The total field at the radar antenna is then approximated as a summation of the backscattered electric fields ($\bar{E}_b$) from individual propagated waves using the Stratton-Chu integral (Fig. 2d). A detailed treatment of the simulation mathematics is presented in Gerekos et al. (2018). Because of higher ice/water dielectric contrast, rays reflecting from a facet within $S_{bed}$ identified as water will have higher amplitude than facets comprised of rock.

The radar pulse repetition frequency ($PRF$) and aircraft velocity $v$ determine the spatial resolution between radar observations in the azimuth direction, $\Delta_a$, as shown in Fig. 2e. In the field, the MARFA instrument employs a native pulse repetition frequency ($PRF$) of 6400 Hz and peak power of 8 kW, then these observations are stacked in post processing to achieve along track resolution of $1m$ before focusing (Peters et al., 2007). We chose to simulate the final (stacked) PRF and power instead of the native parameters for computational efficiency.

## 2.2 Dielectric Material Model

It is instructive to consider energy returned to the radar as the superposition of dielectric and geometric effects. Dielectric effects result from material property changes at an interface, while geometric effects result from the orientation of the interface's topography and the radar antenna. We will begin by discussing the implications of dielectric parameters then move on to geometric considerations.

A material model is applied to the simulation, where all facets on $S_{ice}$ and $S_{bed}$ are assigned a complex dielectric constant, $\epsilon$. The refractive index $\eta$ for each material is derived from $\epsilon$ (Eq. 2). For simplicity, we have chosen a three material model





**Table 1.** Summary of IPR simulation parameters. Parameters were chosen to replicate helicopter-based MARFA ice sounding experiments. Fields marked with an asterisk represent values after post-processing instead of native instrument parameters.

| Parameter | Simulation Value |
|---|---|
| *Geometric* | |
| Aircraft height ($h$) | $500\,m$ |
| Aircraft velocity ($v$) | $30\ m/s$ |
| *Instrument* | |
| Center frequency ($f_c$) | 60 MHz |
| Bandwidth ($B_w$) | 15 MHz |
| PRF* | 30 Hz |
| Power* | 1.71 MW |
| Sampling frequency ($f_s$) | 50 MHz |
| Pulse length ($T_s$) | $1\ \mu s$ |
| Receiving window ($T_r$) | $50\ \mu s$ |

**Table 2.** Complex dielectric constants for materials in radar simulations (Peters et al., 2005; Fujita et al., 2000; Midi et al., 2014; Glover, 2015). For each material, we chose a value within the range published in the literature. Further analysis of sensitivity to these material parameters is presented in the discussion.

| Material | Literature Range $\epsilon$ | Simulated $\epsilon$ |
|---|---|---|
| Ice | $3.18 - 3.2$ | $3.18 + .02i$ |
| Water | $77 - 80$ | $78 + .1i$ |
| Bedrock | $4 - 6$ | $5 + .15i$ |

consisting of ice, rock, and water, with dielectric constants as presented in Table 2. Facets across $S_{ice}$ are assigned $\epsilon_{ice}$. Facets

110 on $S_{bed}$ are assigned $\epsilon_{rock}$ unless they are part of a water structure, which are assigned $\epsilon_{H_2O}$.

For a facet at nadir, absolute reflectivity $R_{abs}$ results from contrast between the refractive indices of ice and the bed material ($\eta_{ice}$ and $\eta_{rock}$, respectively). A horizontal facet assigned $\epsilon_{rock}$ will have a lower reflectivity than one assigned $\epsilon_{H_2O}$ by about $15\,dB$ (Eq. 2). Real IPR instruments are rarely calibrated to measure absolute reflectivity, and thus changes in received power, corrected for ice attenuation loss and geometric scattering, are assumed proportional to relative changes in reflectivity at the bed

115 ($R_{rel}$). This forms the basis for the widely accepted assumption that $R_{rel} \geq 10 - 15\,dB$ over surrounding reflections implies the presence of liquid water (e.g. Young et al. (2016); Peters et al. (2007); Rutishauser et al. (2022)).

$$R_{abs,m} = \left( \frac{\eta_{ice} - \eta_m}{\eta_{ice} + \eta_m} \right)^2, \qquad \eta_m = \sqrt{Re(\epsilon_m)}, \qquad m = rock, H_2O \tag{2}$$





The simulations presented in this paper assume only the three materials described in Table 2, with well defined boundaries between hydrological and bedrock features. In real-world sub-glacial environments, additional material heterogeneity such as clays or hydrated tills exist, in addition to ambiguity in hydrological boundaries. We address the implications of this relatively simplistic material model in the discussion section.

## 2.3 Geometric Model

We ultimately seek to develop the Stratton-Chu simulation method for constraining the extent and cross-sectional geometry of sub-glacial water features. Given this objective, it imperative to consider a menu of geometric constraints and how simulation parameters will emulate real-world IPR returns from different targets. A basic, hypothetical simulation geometry is shown in Fig. 3a. Simulations consist of hypothetical flat surfaces $S_{ice}$ at elevation $0m$ and $S_{bed}$ at $-d_{ice}$, where $d_{ice}$ is the nominal ice thickness in $m$. The flight path is defined along the $y$-direction, with the radar's dipole antenna oriented along the $x$-direction. On the bed surface, a straight channel of width $c_w$ is oriented perpendicular to the flight path.

### 2.3.1 Surface Roughness

Small-scale topography variation below the radar's detection limit, which we will refer to as roughness, can impact reflectivity by diffusely scattering incident radar energy. To account for this effect, random isotropic Gaussian variation is introduced to both $S_{ice}$ and $S_{bed}$ via Eq. 3. $l_c$ is the correlation length in both the $x$ and $y$ directions. To capture scattering behavior due to topography changes at the radar wavelength scale, $l_c$ should be at least a few times the facet length $l_f$, while remaining near $\lambda$. In our simulations, both $l_f$ and $\lambda$ are $\sim 5m$, leading us to select $l_c = 15m$.

$$cov_{x,y} = \sigma_m e^{-(x^2+y^2)/l_c^2}, \qquad m = ice, bed \tag{3}$$

When we assume our surfaces are smooth relative to $\lambda$, we must add at least a negligible, non-zero roughness to avoid simulation artifacts, such as Bragg resonance (Gerekos et al., 2020). We always make this assumptions for $S_{ice}$, therefore $\sigma_{ice} = 0.2m$ in all simulations presented here. We also use $\sigma_{bed} = 0.2m$ in "smooth" bed simulations.

Hubbard et al. (2000) measured topography of recently deglaciated bedrock at high resolution in the Swiss Alps. They showed variations near $1m$ over horizontal distances of $15m$, which serves as our upper bound on $\sigma_{bed}$. We acknowledge the Alpine and Thwaites sub-glacial environments may differ considerably. However, roughness studies from the Thwaites region integrate over long horizontal scales irrelevant to radar scattering (Bingham and Siegert, 2009; Hoffman et al., 2022). We proceed with the understanding that our range for $\sigma_{bed}$ from $0.2m$ to $1m$ may represent an imperfect but reasonable range of expected variation in small-scale topography.

### 2.3.2 Simulation Radius

An appropriate choice of simulation radius $R$ is vital to accurately simulate radar geometric scattering from the sub-glacial environment. $R$ defines the radius of a vertical cylinder, bounding the simulation scope at each aircraft position increment. If





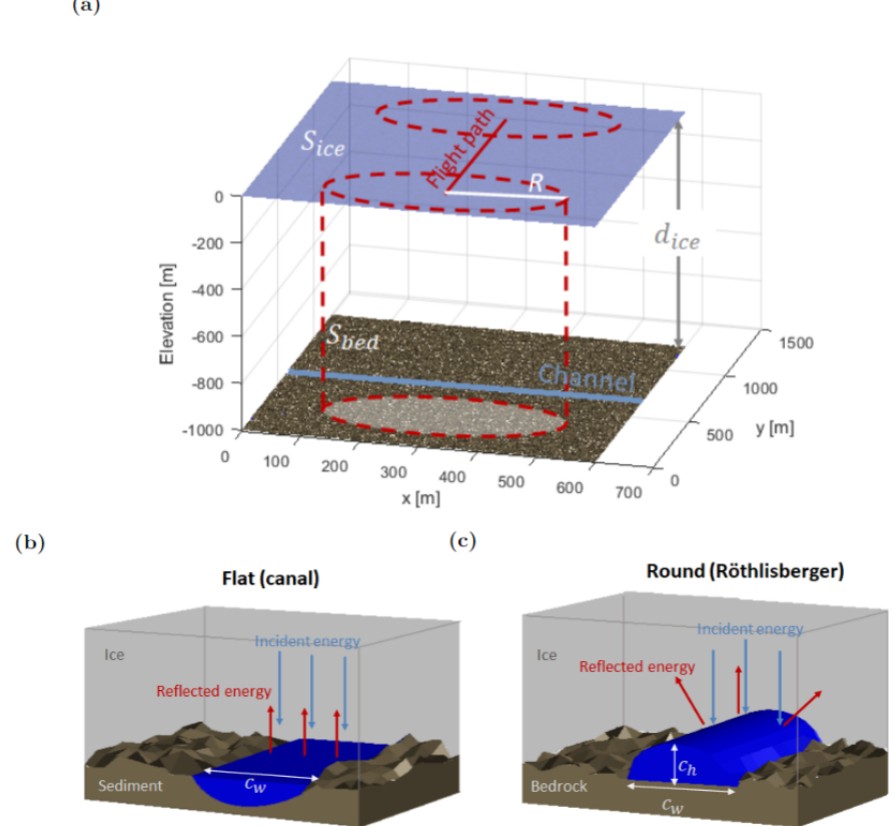

**Figure 3.** Radar simulation geometry a) Flight path (red), aircraft / channel orientation, simulation radius ($R$) across model footprint. b) A flat canal with width $c_w$. c) Geometry of a Röthlisberger or round channel with width $c_w$.

$R$ is sufficiently large to capture the entire antenna beam pattern cast on the bed surface, then the simulation will provide a complete representation of off-nadir clutter and target range migration. In many layered radar simulation experiments, $R$ is

chosen primarily to capture off-nadir clutter at the sub-surface target's apparent depth (Gerekos et al., 2018). Our experiments involve both a thick ice material layer and smooth $S_{ice}$ relative to $\lambda$, limiting the impact of off-nadir clutter. Therefore, we base our choice for $R$ on two alternative criteria:

- $R$ must be greater than pulse limited radius $R_{pl}$ at the glacier bed.

- $R$ captures adequate range migration to facilitate along track focusing.

The first criterion assumes the majority of returned energy from a nadir-directed radar will come from within the pulse-limited footprint ($R_{pl}$) beneath the aircraft. Eq. 4 approximates an upper limit on $R_{pl}$ for a radar instrument with bandwidth $B_w$, where $c$ is the speed of light in a vacuum. Actual $R_{pl}$ will always be smaller than Eq. 4 predicts, due to refraction at



the air/ice interface. For our simulations, with $d_{ice} \sim 1000m$, $R_{pl} \approx 130m$ (Eq. 4). We consider this the minimum acceptable simulation radius, although clearly changes in aircraft height or simulated ice thickness will alter this limit.

$$R > R_{pl} \approx \sqrt{\frac{c(d_{ice} + h)}{B_w \eta_{ice}}} \qquad (4)$$

An appropriate choice for $R$ must also consider the desired range cell migration ($RCM$) at the bed surface. $RCM$ is proportional to the change in physical distance a signal travels through air ($r_{air}$) and ice ($r_{ice}$) to reach a target as the radar moves past (Eq. 5). In Synthetic Aperture Radar (SAR) processing, an aperture length $L_a$ is chosen with sufficient range cell migration to optimize along-track focusing. This process improves signal-to-noise ratio and along-track resolution (Cumming

and Wong, 2005). In order to facilitate simulated data focusing, $R$ must be greater than the aperture required for desired range migration. Selection of $L_a$ and the focusing process is described at length in *Simulated Data Processing*.

$$RCM(y) = \frac{2f_s}{c} \left[ (r_{air}(y) - h) + \eta_{ice}(r_{ice}(y) - d_{ice}) \right] \qquad (5)$$

Our simulations target 3 cells of range migration at the bed surface ($RCM(L_a) \geq 3$). For $d_{ice} = 1000m$, and sampling frequency $f_s = 50MHz$, this translates to $L_a = 277m$. All simulations presented here use $R = 300m$, meeting both the pulse-

limited and range migration criteria. Real airborne IPR focusing typically includes more range cell migration ($RCM(L_a) = 5$ fast-time samples for MARFA), however such a simulation radius would be computationally unrealistic. Therefore, our choice of $R$ represents a compromise, and the implications will be discussed in the *Results and Discussion* section. Simulations in thicker ice may require significantly larger $R$ or further compromises in range migration.

### 2.3.3 Channel Geometry

We seek to distinguish between two channel geometries common to sub-glacial hydrology. A canal-like structure (Fig. 3b) will have a flat cross-section and produce a specular reflection. This type of feature is common when the surrounding bed is comprised of sediment or other soft material, and the water pressure is high. Conversely, in a Röthlisberger channel (Fig. 3c), sub-glacial water carves a path through the ice above the bed surface (Röthlisberger (1972)). This type of channel is likely to form in areas where the substrate is impermeable bedrock with low water pressure (Walder and Fowler, 1994; Schroeder et al.,

2013). We have confined ourselves to Röthlisberger channels and flat canals in this study as examples of common large-scale hydrological structures. Other known hydrologic features, such as Nye channels, may appear radiometrically similar to small canals, depending on their size and the radar wavelength. As algorithms and computational power enable higher resolution models, additional sub-glacial structures are a logical extension of this work.

A Röthlisberger channel has an elliptical cross section, with height $c_h$ in addition to channel width $c_w$. We can infer from

basic geometry it will reflect radar energy divergently (Fig. 3c), therefore the actual radar signature of a sub-glacial channel will be the superposition of its geometry and dielectric contrast. When representing the Röthlisberger curvature with flat facets, we must approximate the divergent scattering behavior by capturing reflections from multiple facets in the channel cross-section





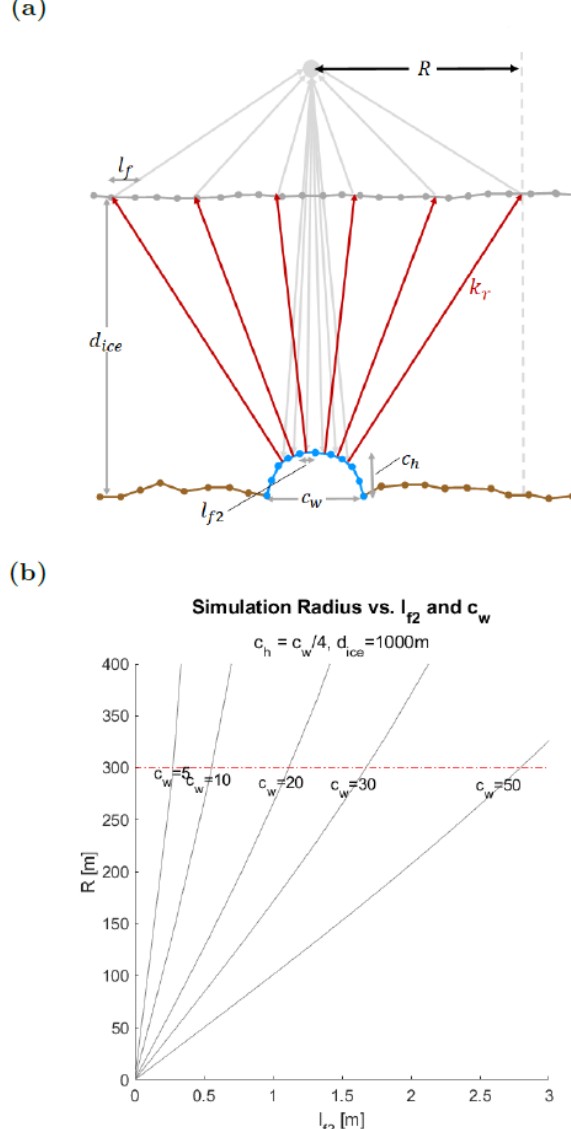

**Figure 4.** a) Schematic of simulation geometry for an elliptical cross-section of the Röthlisberger channel at nadir. A second, smaller facet length scale $l_{f2}$ is introduced to facilitate accurate representation of curvature and divergent reflection character. b) Relationship between $R$ and $l_{f2}$ is defined for a range of $c_w$. We seek to limit $R \leq 300m$ for computational efficiency, imposing an upper limit on $l_{f2}$ when simulating Röthlisberger channels

at nadir. In our simulations, we seek to capture reflections from at least the upper 6 facets within the simulation footprint as a reasonable estimate of Röthlisberger channel scattering (Fig. 4a).





Given our desired range of $c_w$ and the constraint already imposed on $R$, it is clear that $l_f = 5m$ is too large to capture the

scattering effect of a Röthlisberger channel. For our smallest channels ($c_w = 5m$), $l_f \leq .25m$ is required to capture the desired

facet reflections (Fig. 4b). Setting such a high resolution over all of $S_{ice}$ and $S_{bed}$ is computationally unrealistic and negates

many of the benefits of the Stratton-Chu simulation method (Gerekos et al., 2018). Therefore, we introduce a second facet

length scale, $l_{f2}$, which defines the facet length on $S_{ice}$ and $S_{bed}$ only in a Röthlisberger channel location. $l_{f2}$ has a maximum

of $\sim c_w/20$, for $d_{ice} = 1000m$ and $c_h = c_w/4$ (Fig. 4b). When simulating flat canals, higher resolution channel facets are

unnecessary due to the specular nature of the reflection, and therefore $l_{f2} = l_f$.

## 2.4    Simulated Data Processing

The simulator outputs rangelines, which represent the electric field strength vs. fast-time $\tau$ returned from a single pulse, at

azimuth time $a$. $\tau$ is discretized into range bins (index $j$) with increments of $1/f_s$, where $f_s$ is the sampling frequency. The

rangelines are compiled sequentially in azimuth, or slow-time, into a 2-D raw radargram matrix ($\xi_{raw}$). Azimuth increments

(index $k$) are equally spaced in time at $1/PRF$.

    The rangelines are then focused using a version of the Range Doppler Algorithm (RDA) (Cumming and Wong, 2005;

Hélière et al., 2007). We first perform range compression by convolving each raw rangeline with the complex conjugate of the

radar chirp, $g(\tau)$ over the radar receiving window $T_r$. The operation is performed by multiplication in the fast-time frequency

domain.

$$g(\tau) = e^{-i\pi B_w \tau^2/T_s} \tag{6a}$$

$$\xi_{RC}(a,\tau) = \int_0^{T_r} \xi_{raw}(a,\tau+\tau')\,g(\tau)^* d\tau' \tag{6b}$$

We produce the focused radargram ($\xi_f$) by convolving the range compressed radargram ($\xi_{RC}$) with a 1-D reference function

($\phi$) in along-track blocks (Eq. 7). The block size, $L_a$, for each fast-time value of $\tau$ is chosen such that range migration equals 3

fast-time samples. Thus it is important to note that $L_a$ increases with depth, and simulation radius $R$ must be greater than the

maximum anticipated $L_a$.

$$\xi_f(a,\tau) = \int_{-L_a/2}^{L_a/2} \xi_{RC}(a+a',\tau)\,\phi(a',\tau)^* da' \tag{7}$$

A deeper discussion of mathematics and block processing required for along-track focusing is presented in the *Appendix A1*.

    In all radargrams, we convert fast-time to physical depth ($d$) and slow time to along-track distance ($y$) via Eqs. 8 and 9,

where $\tau_s = 2h/c$ is the fast-time value for the surface reflection, and $PRF$ is the radar pulse repetition frequency.





$$d_j = \frac{c}{2\eta_{ice}}(\tau_j - \tau_s), \quad \eta_{ice} = \sqrt{Re(\epsilon_{ice})} \tag{8}$$

$$y_k = k\frac{v}{PRF} \tag{9}$$

The focused field in $\xi_f$ is converted to power in decibels. We define along-track bed reflectivity in absolute terms ($R_{abs}$) by taking the maximum reflected power beneath the ice surface (Eq. 10a). Relative reflectivity ($R_{rel}$) compares $R_{abs}$ to the mean reflectivity for a simulation with the same $\sigma_{bed}$ and $\sigma_{ice}$, but no channel present ($R_{abs,0}$). $R_{rel}$ therefore measures the relative reflectivity gain observed by the radar due to the channel's presence vs. surrounding frozen bed material.

$$R_{abs}(y_k) = max\{RG_{f,dB}(d,y_k)\}, \quad d > 0 \tag{10a}$$


$$R_{rel}(y_k) = R_{abs}(y_k) - mean\{R_{abs,0}\} \tag{10b}$$

## 2.5   Hypothetical Simulations

A basic simulation geometry is shown in Fig. 3. Simulations consisted of hypothetical flat surfaces $S_{ice}$ and $S_{bed}$ with elevations
of $0\,m$ and $-1000\,m$, respectively. Each surface had isotropic Gaussian roughness as defined above. On the bed surface a single channel of width $c_w$ was oriented perpendicular to the flight path. Channel cross-sections were either flat canal-like structures (Fig. 3b), or round Röthlisberger channels with channel height of $c_w/4$ (Fig. 3c).

A series of simulation experiments were run for both types of channels. Channel width, and basal roughness were varied according to Table 3. Simulations involving flat canals used a single facet length of $5\,m$, while Röthlisberger channels necessi-
tated a smaller $l_{f2}$ in the channel location to accurately represent geometric scattering. Rangelines from each simulation were processed as described above, and along track bed reflectivity from the focused radargram $R_{rel}$ was compared for various scenarios.

## 2.6   Application to Real-World Hypothesis Testing, Thwaites Glacier

To demonstrate the simulator's utility in sub-glacial hypothesis testing, we compared simulated reflectivity to a single flight
line conducted with UTIG's 60 MHz MARFA instrument. The $16\,km$ line (THW2/ UBH0c/ X243a) was part of a 2022 airborne radar survey of Thwaites Glacier, employing an AS-350 B2 helicopter at a nominal height of $500\,m$ above ground level and target velocity of $30\,m/s$. Precise aircraft positioning and orientation were recorded with an onboard Renishaw laser altimeter, Trimble Net-R9 dual frequency GNSS, and Novatel SPAN IGM-1A inertial navigation, as described in Lindzey et al. (2020).



**Table 3.** Summary of IPR simulation parameters for each channel geometry.

| Parameter | All Simulations |
|---|---|
| Nominal ice thickness($d_{ice}$) | $1000\,m$ |
| Simulation Radius ($R$) | $300\,m$ |
| Facet length ($l_f$) | $5\,m$ |
| Range migration for SAR Aperture ($RCM$) | $3\,$cells |
| Correlation length ($l_c$) | $15\,m$ |
| Surface roughness ($\sigma_{ice}$) | $0.2\,m$ |
| Basal roughness ($\sigma_{bed}$) | $0.2, 1\,m$ |
| Channel width ($c_w$) | $0, 5, 10, 20, 30, 50\,m$ |

| Parameter | Flat Canals | Round Channels |
|---|---|---|
| Channel facet size ($l_{f2}$) | $5\,m$ | $.25 - 2.5\,m$ |
| Channel height($c_h$) | $0$ | $c_w/4$ |

THW2/ UBH0c/ X243a transects one of the proposed Hager et al. (2022) channel routes, and the radargram shows an isolated

bright reflection coincident with this location, as shown in Fig. 5. Data from THW2/ UBH0c/ X243a were range compressed, corrected for geometric spreading loss and aircraft position, and focused in azimuth as described in Peters et al. (2007). This azimuth focusing is analogous to the procedure described in the *Simulated Data Processing* section, although a longer aperture sufficient for range migration of 5 cells is used.

The along-track surface and bed profiles were picked from the radargram. One-way attenuation loss of $13.8\,dB/km \pm$

$1.4\,dB/km$ was estimated using a spatially constrained linear regression model as outlined in Schroeder et al. (2016).

We simulated a $4\,km$ segment of THW2/ UBH0c/ X243a containing the proposed channel, as depicted in Fig. 5b. Along-track ice and surface elevations at nadir were calculated from the THW2/ UBH0c/ X243a focused radargram. From these data, we built elevation matrices for both the ice and bed surfaces ($S_{ice}^{rad}, S_{bed}^{rad}$) which vary in $y$ according to the respective radar profiles, but have no $x$ variation.

To build appropriate across-track ($x$) topography, separate 2-D topographic matrices were created from BedMachine V2 data ($S_{ice}^{BM}, S_{bed}^{BM}$) (Morlighem, 2020). The radar and BedMachine derived topography were superimposed to create simulation surfaces $S_{ice}$ and $S_{bed}$ via Eq. 11, where $w$ is a quadratic weighting function varying between 1 at the surface edges and 0 at nadir. $S^{rough}$ is the appropriate isotropic Gaussian surface roughness as described above.

$$S = wS^{BM} + (1-w)S^{rad} + S^{rough} - z_0(y) \tag{11}$$



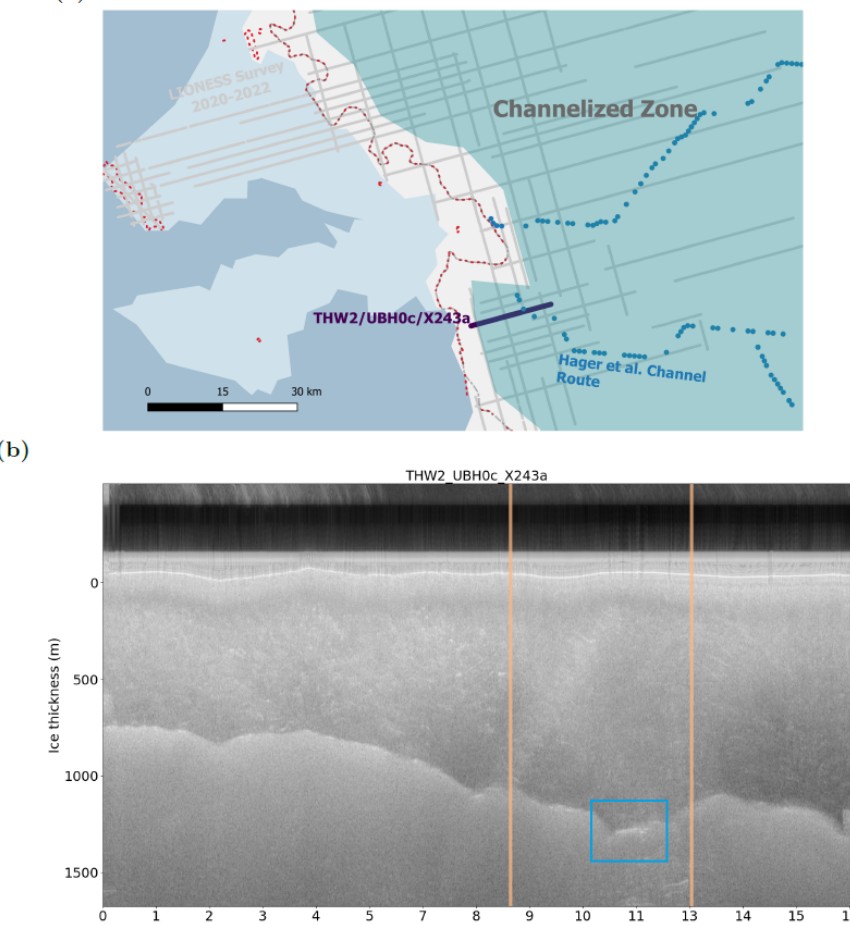

**Figure 5.** a) Region near Thwaites central grounding line (red dots), with flight lines from 2020-2022 LIONESS survey shown in grey. The Hager et al. (2022) channel locations and flight line THW2/ UBH0c/ X243a are highlighted. b) Focused radargram for THW2/ UBH0c/ X243a. The bright bed reflection corresponding to the proposed channel location is boxed. Vertical lines represent extent of the flight line re-produced in simulations.

In an IPR radar survey, the aircraft attempts to "drape" the ice surface by flying at a constant height above ground level ($500m$ for UTIG helicopter based surveys). We simulate this with a polynomial interpolation of the radar-derived ice elevation along track, $z_0(y)$. For THW2/ UBH0c/ X243a, $z_0$ is a 7th order polynomial, although in practice the polynomial order is somewhat subjective. The fitted function should approximate major terrain features in $S_{ice}^{rad}$ with gentle elevation changes, consistent with actual aircraft operation.



$z_0(y)$ is subtracted from $S_{ice}$ and $S_{bed}$, setting the average ice surface elevation to $\sim 0\,m$ (Eq. 11). The simulated aircraft elevation $h$ is a constant $500\,m$. This approach preserves known ice geometry at nadir, and minor topographic features appear as variations in aircraft range to target.

The dielectric material model was applied to $S_{ice}$ and $S_{bed}$ as described above. We ran individual simulations using the same surfaces, varying the width and geometry of across-track oriented channels in the location identified in Hager et al. (2022).

Relative bed reflectivity from each simulation result was compared to the actual relative reflectivity from the focused THW2/ UBH0c/ X243a radargram (Fig. 8c). The comparison between simulated and real data provides constraints on the extent and geometry of any real hydrological features at this location, as we examine in the *Results and Discussion* section.

## 3   Results and Discussion

### 3.1   Geometric Effects

Basal roughness ($\sigma_{bed}$) impacts the absolute reflectivity of the solid bed material in our simulations ($R_{abs,0}$), which we use as a baseline for calculating relative impact of channels on the radar echo (Eq. 10b). Simulations excluding liquid water, with $\sigma_{bed}$ of $0.2\,m$ and $1\,m$ produced mean $R_{abs,0}$ equal to $-107.9 \pm 2.5\,dB$ and $-114.1 \pm 2.4\,dB$, respectively. These results are intuitive, as we expect scattering loss from a rougher surface to be larger and more variable, resulting in lower absolute reflectivity. It is instructive to quantify this intuition.

Our simulated results indicate that small scale roughness change at the bed may significantly alter radar echoes even without a change in dielectric properties. As frozen till or sediment replaces rough bedrock, for example, the resulting radar echo power may theoretically increase by $6.2\,dB$. This indicates that IPR detection of sub-glacial water must be more nuanced than assuming large changes in $R_{rel}$ constitute a liquid water signature. Spatial heterogeneity in basal material, roughness, and topography could produce significant reflectivity changes. Therefore, IPR analysis of sub-glacial hydrological systems should

consider $R_{rel}$ changes in the context of local topography and the likelihood of substrate transitions.

Hypothetical simulations of Röthlisberger channels and flat canals with the same $c_w$ demonstrate the distinctly different radar signatures expected for the two geometries. Figure 6 shows along-track relative reflectivity for the two channel types, with $c_w = 20\,m$, compared to a frozen substrate with no liquid water. Flat canals have a distinct peak centered near the canal location, with maximum $R_{rel} = 19.5\,dB$ when $\sigma_{bed} = 0.2\,m$ and $22.9\,dB$ when $\sigma_{bed} = 1\,m$. Regardless of roughness, a $20\,m$

flat canal influences the radar echo for a few hundred meters along-track. The reduced gain of $3.4\,dB$ for a canal surrounded by a smoother substrate is consistent with the higher absolute reflectivity of the surrounding surface as discussed above.

Röthlisberger channels will scatter energy divergently (Fig. 3c), therefore we anticipate a smaller magnitude impact to $R_{rel}$ over a greater distance than an equivalent flat canal. When flowing through a rough bed (Fig. 6b), a $20\,m$ simulated Röthlisberger channel increased along-track $R_{rel}$ for more than $500\,m$. The maximum peak in $R_{rel}$ for this channel is $10.3\,dB$,

and occurring when the channel is far from nadir. In our smoother bed simulations, the same $20\,m$ Röthlisberger cross-section produced peak $R_{rel}$ of only $7.6\,dB$ (Fig. 6a), which could be nearly indistinguishable from fluctuations in reflectivity from the



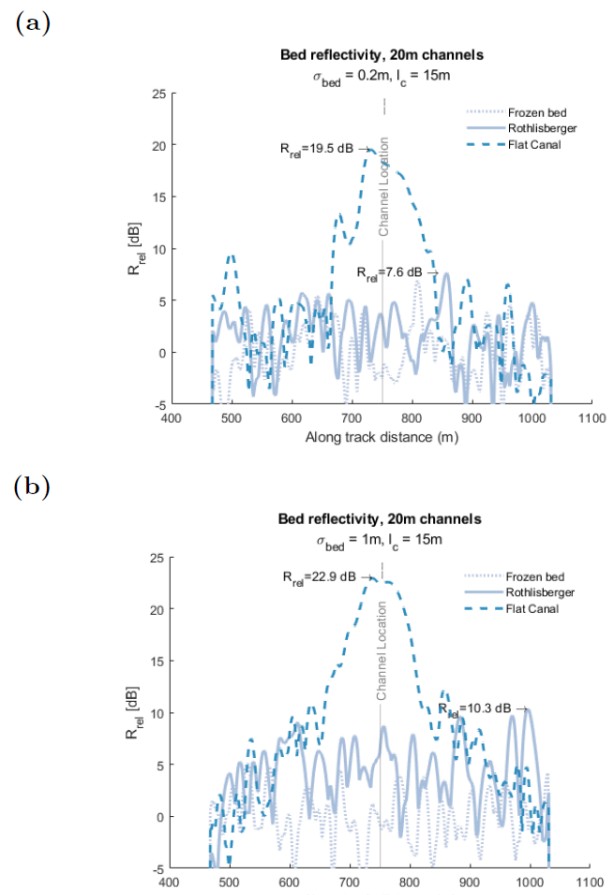

**Figure 6.** Simulated along track $R_{rel}$ for 20m channels compared to a dry substrate with a) $\sigma_{bed} = 0.2m$. b) $\sigma_{bed} = 1m$. All simulations had same Gaussian roughness correlation length $l_c = 15m$.

frozen substrate. Larger Röthlisberger channels produce only moderate gains in $R_{rel}$, with the largest $(50m)$ channels having peak $R_{rel} = 13.2\,dB$ (Fig. 7).

These results imply that a real world Röthlisberger channel, even one of significant size, may not have an obvious reflectivity increase. More advanced analysis techniques may be required to detect such channels, such as examination of specularity content as described in Schroeder et al. (2015). Generating specularity content requires longer along-track focusing apertures at significant computational expense. Therefore, we leave simulations of specularity content to future work employing higher powered computing resources, or incorporating simulated facet roughness for additional realism and efficiency (Gerekos et al., 2023).

Figure 7 shows peak $R_{rel}$ generally increases with channel width, as an increasing proportion of the area within the radar's footprint contains liquid water. Flat canals exhibit a much stronger correlation between radar reflectivity response and $c_w$ than





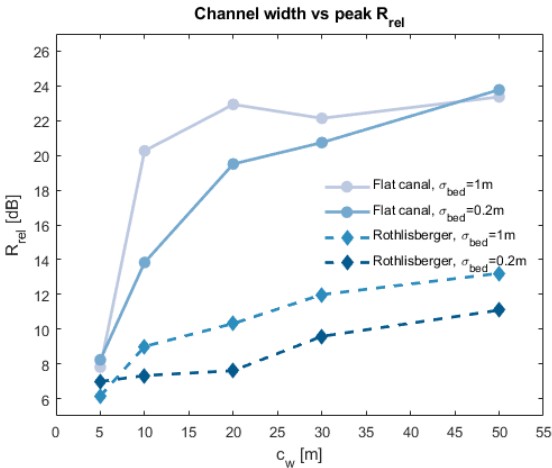

**Figure 7.** Positive correlation between $c_w$ and $R_{rel}$ for both Röthlisberger channels and flat canals. The increase in $R_{rel}$ is much more pronounced for flat canals.

Röthlisberger channels when $c_w < 20m$. The simulator also demonstrates that peak $R_{rel} = 15\,dB$ might be achieved when only $\sim 4 - 6\%$ of the radar footprint contains liquid water in a flat canal (Fig. 7, *Flat canal*).

As bed roughness increases, frozen areas scatter radar energy diffusely, increasing contrast between the hydrological feature and surrounding material. The effect is most pronounced for flat canals with $c_w = 10 - 20m$. The impact of the bed roughness in flat canals diminishes as $c_w$ exceeds $30m$, when reflected energy from the water feature dominates the radar return.

## 3.2 Dieletric Uncertainty

When considering the full range of literature values listed in Table 2, Eq. 2 indicates that liquid water increases dielectric reflectivity by $12.5 - 21.5\,dB$ vs. unfrozen bedrock. $\sim 95\%$ of that variation comes from uncertainty in the primary substrate material, $\epsilon_{rock}$. The simulations presented in this paper assume a simple three material model (Table 2), with well defined boundaries between hydrological and bedrock features. Our choice in bedrock dielectric constant could introduce error in $R_{rel}$ up to $\sim 6\,dB$ in our real-world application to Thwaites Glacier. This uncertainty could be reduced with additional information about the local geology, or if the radar acquisition system has been calibrated over targets of known dielectric contrast. Such a calibration is often unrealistic for real IPR survey conditions, but when possible would allow the use of absolute instead of relative reflectivity and enable better inferences about the dielectric properties of the bed material.

Given our short simulation distances of only a few $km$, we did not examine the impact of spatial variations in geology on simulation results. We also did not consider the presence of frozen tills ($\epsilon \sim 2.7$), as our chosen real-world scenario near the grounding line of Thwaites Glacier is unlikely to contain such a substrate. Hydrated tills with $\epsilon \sim 18$ (Peters et al., 2005) would be possible in this environment, though we did not introduce such a bed material here for simplicity. Each of these permutations should be considered in future work as appropriate for the environmental conditions being simulated.



### 3.3 THW2/ UBH0c/ X243a Simulations

Figure 8 compares a $4\,km$ segment from THW2/ UBH0c/ X243a IPR data with a simulation containing no water and $\sigma_{bed} = 0.2\,m$. There are two bright reflections in this section of the radargram (Fig. 8a). The first is centered around $1100\,m$ along-track with a maximum $R_{rel} = 11.6\,dB$. The second is a broad area of high reflectivity between $2300 - 2900\,m$, with $R_{rel}$

peaks ranging from $13$ to $15\,dB$ (Fig. 8c). This reflector coincides with the location of persistent Röthlisberger channelization proposed in Hager et al. (2022), and is the primary area of interest along THW2/ UBH0c/ X243a for this study.

    The radargram from our frozen bed simulation (Fig. 8b) captures the basic bed topography well, but along-track reflectivity is not always aligned with the real THW2/ UBH0c/ X243a radargram. Simulated reflectivity near $1100\,m$ is consistent with the real data, indicating that this reflectivity peak could be the result of geometric effects from topography, rather than dielectric

contrast from a water feature. However, for the region between $2300 - 2900\,m$, mean simulated $R_{rel}$ is $22.8\,dB$ below the value observed in the real THW2/ UBH0c/ X243a data (Fig. 11). In our material model, a gain of this magnitude could only be consistent with a change in dielectric properties from rock to liquid water. It is also unlikely to be a Röthlisberger channel, since we have demonstrated such geometry is not conducive to increased reflectivity of more than $13.2\,dB$ for very large channels. We therefore assume that this basal reflector at $2300 - 2900\,m$ along-track must represent a specular hydrological structure,

such as a flat canal, of unknown dimensions.

    It is also important to note that the simulation in Fig. 8d exhibits along-track $R_{rel}$ variation $> 30\,dB$ without any change in dielectric properties at the bed. This demonstrates that significant changes in bed echo strength are possible due to topography alone. When inferring the presence of sub-glacial hydrological features from IPR data, care must be taken to consider reflectivity within the context of bed topography, hydraulic potential, attenuation, and other factors which may influence the radar

echo strength. This observation also demonstrates the value of our simulation methodology for confirming the presence and extent of sub-glacial water.

    When a single flat canal with $c_w = 20\,m$ was added to the simulation, we observe a peak $R_{rel}$ of $6.8\,dB$ over a narrow $150\,m$ range along-track (Fig. 9). This reflectivity gain is consistent with our findings from hypothetical simulations described above. However, the gain in reflectivity and along-track extent are insufficient to match the reflectivity profile observed in the real

THW2/ UBH0c/ X243a IPR data (Fig. 9b, Fig. 8c). Therefore, we conclude that the reflector cannot be a narrow, isolated flat canal.

    Based on the above results, we hypothesize the reflector between $2300 - 2900\,m$ must be a distributed hydrological feature, such as a broad area with multiple flat canals. This hypothesis is compatible with the topographic context, given that it is in a low-lying area with steep topography just down-glacier. This is an ideal location for till and liquid water to accumulate if the

ice exists at its pressure melting point.

    Figure 11 compares the difference between THW2/ UBH0c/ X243a and simulated $R_{rel}$ for five simulations of distributed hydrological features. We compare both the mean and peak difference in $R_{rel}$ between $2300 - 2900\,m$, and conclude that each more closely approximates THW2/ UBH0c/ X243a than our frozen bed or single $20\,m$ channel simulation. First, we consider the glacier sliding condition originally proposed by Weertman (1964). In this scenario, a thin water film (perhaps a



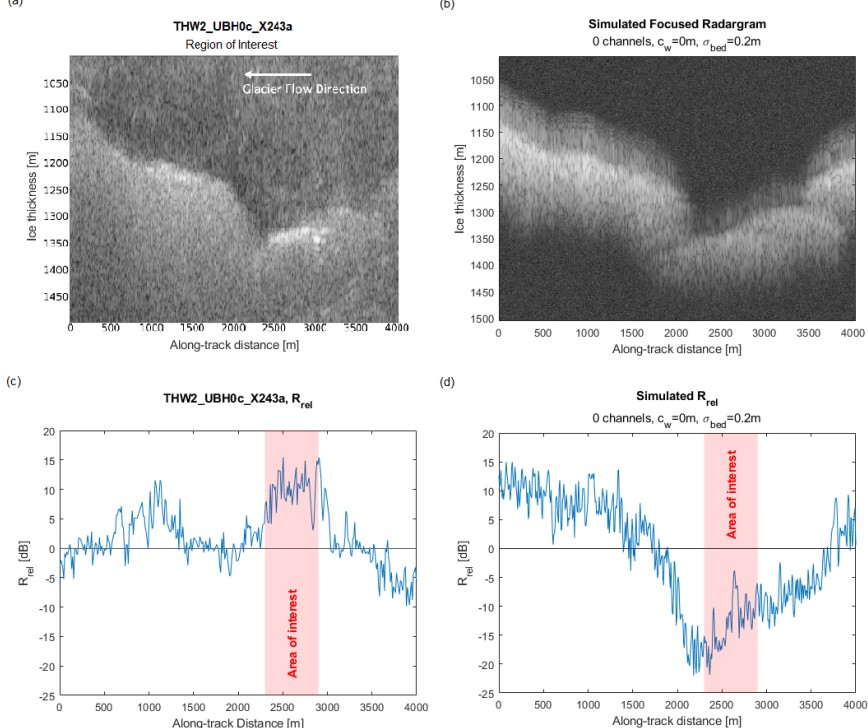

**Figure 8.** a) A $4km$ region from actual THW2/ UBH0c/ X243a radargram containing the Hager et al. (2022) channel location. b) Simulated THW2/ UBH0c/ X243a radargram with no water, $\sigma_{bed} = 0.2m$. c) Bed relative reflectivity from the actual THW2/ UBH0c/ X243a. The proposed channel location is highlighted in red as the area of interest. d) Simulated THW2/ UBH0c/ X243a relative reflectivity with no water, $\sigma_{bed} = 0.2m$. The along-track location corresponding to the area of interest is highlighted in red.

few centimeters thick) coats the basal interface. We model this by maintaining the basal topography with $\sigma_{bed} = 0.2m$, but all facets from $2300m \leq y \leq 2900m$ are assigned $\epsilon_{H_2O}$. This simulation has a broad increase in reflectivity in the area of interest (Fig. 10a), however the mean $R_{rel}$ between $2300 - 2900m$ is $9.4\,dB$ below the actual THW2/ UBH0c/ X243a value (Fig. 11).

Figure 10b shows simulated $R_{rel}$ when 30 narrow, flat canals ($c_w = 10m$) were placed over the same $600m$ region along-track. This creates a region where $50\%$ of the area is covered with $10m$ flat canals between areas of bed material with smooth

basal roughness ($\sigma_{bed} = 0.2m$). Mean $R_{rel}$ at $2300 - 2900m$ for this simulation is slightly higher than the Weertman film scenario in Fig. 10a, but still $8.2\,dB$ below the target $R_{rel}$ from THW2/ UBH0c/ X243a.

The third simulation in Fig. 10c has 8 larger flat canals ($c_w = 30m$) evenly spaced across the same $600m$ region. This simulation has less water coverage than the $30x10m$ flat canal simulation ($40\%$ vs. $50\%$), yet the wider channels increased mean $R_{rel}$ by $3.5\,dB$ (Fig. 11). This result reinforces that the shape of hydrological feature, not just the extent, makes a

significant difference to the resulting reflectivity profile. Mean $R_{rel}$ for the $8x30m$ simulation was $4.8\,dB$ below actual. When the geometry is changed to 6 canals of $50m$ (Fig. 10d), mean $R_{rel}$ improves to just $2.3\,dB$ below THW2/ UBH0c/ X243a.



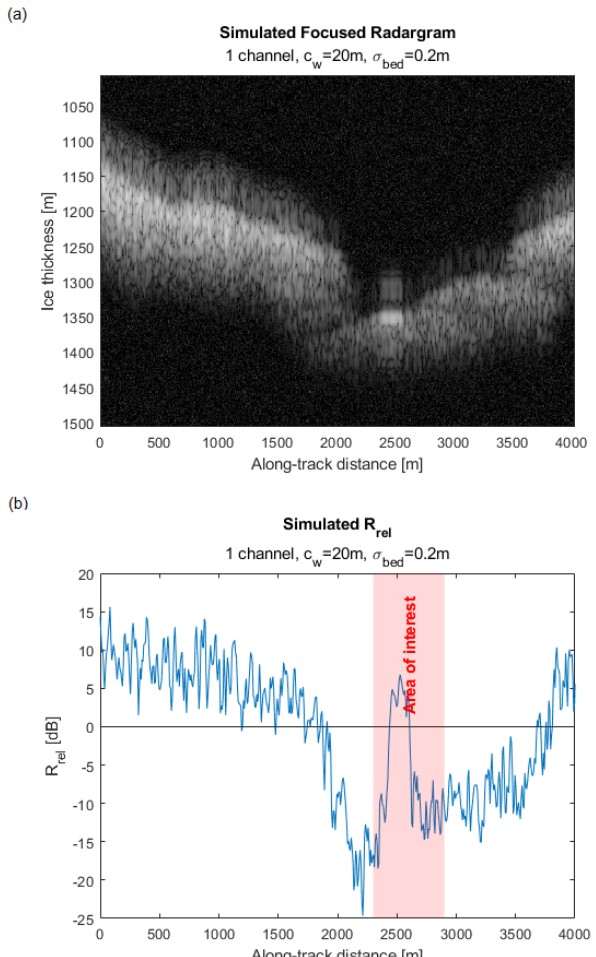

**Figure 9.** a) Simulated radargram with a single 20m canal at the bed. b) Relative reflectivity for simulation with 20m canal at the bed.

The final simulation includes a very broad area of specular ($\sigma_{bed} = 0m$) water covering the bed between $2300 - 2900m$. Due to its size, we refer to this feature as the "$600m$ lake" in Figs. 10 and 11. Mean $R_{rel}$ over $2300 - 2900m$ for this simulation deviates by only $0.6\,dB$ from the actual THW2/ UBH0c/ X243a data. However, this $600m$ lake simulation includes several

peaks as high as $18\,dB$, which is $3\,dB$ higher than the maximum peaks observed in THW2/ UBH0c/ X243a. The maximum $R_{rel}$ from both the $8x30m$ and $6x50m$ simulations more closely matched the peaks observed in THW2/ UBH0c/ X243a.

Based on our simulation results, we infer our hydrological feature at $2300 - 2900m$ is a wide area of distributed water. The simulation results leave some ambiguity as to the precise canal width, but they likely average at least $\sim 30 - 50m$. The area is probably covered by at least $50\%$ water, as $100\%$ coverage may induce higher $R_{rel}$ than was actually observed in THW2/

UBH0c/ X243a.





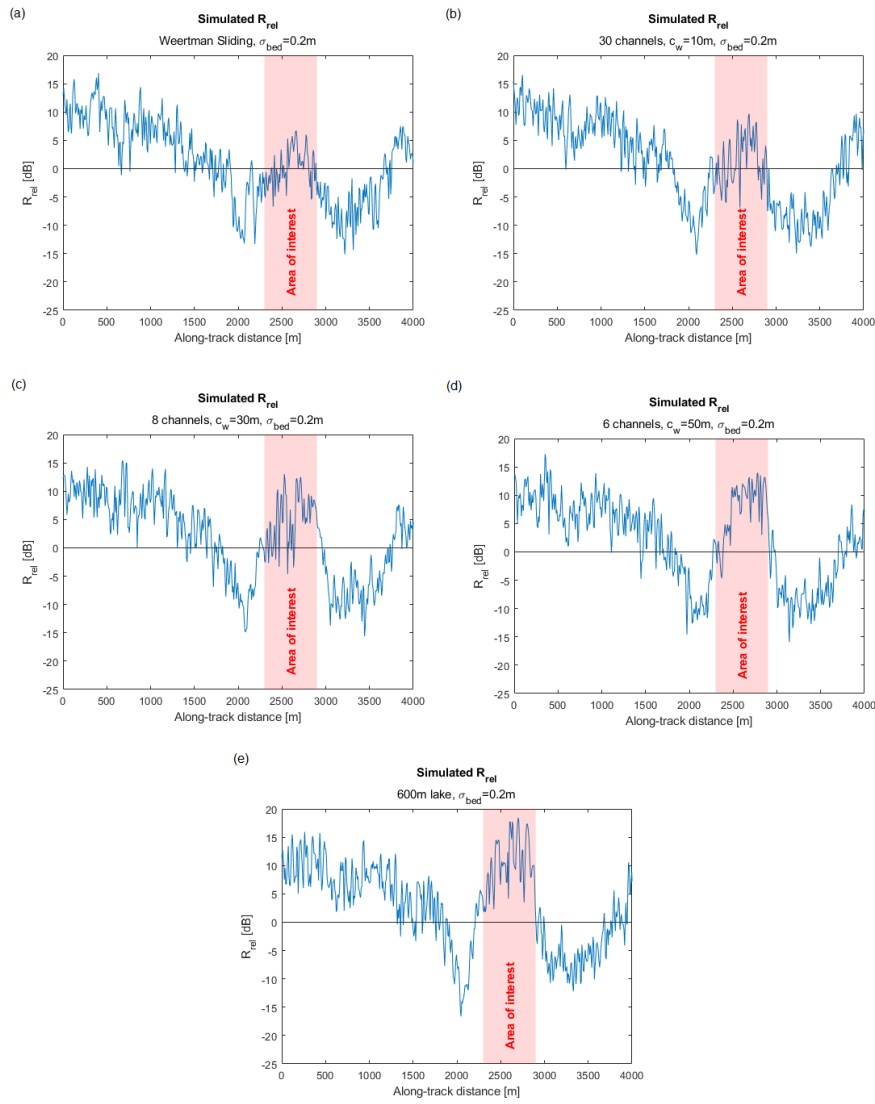

**Figure 10.** $R_{rel}$ for distributed hydrology simulations at $2300 - 2900\,m$ along-track. a) a $600m$ wet bed surface (Weertman sliding), b) 30 flat canals with $c_w = 10m$, c) 8 flat canals with $c_w = 30m$, d) 6 flat canals with $c_w = 50m$, e) the entire $600m$ area covered with a flat, specular water body (e.g. sub-glacial lake).

Several notable inconsistencies between our best simulations and the original THW2/ UBH0c/ X243a data remain. First, $R_{rel}$ over the beginning and final $500m$ of all the simulations are about $\sim 5 - 8\,dB$ higher than the THW2/ UBH0c/ X243a IPR data. This may indicate a change in $\sigma_{bed}$ near the edges of the simulated region. The low-lying area in the region is a perfect topographical feature to accumulate silt and clay deposits, which are likely have low intermediate-scale roughness. The

steeper and elevated topography near the edges of the simulated region may be exposed bedrock, with larger roughness, which



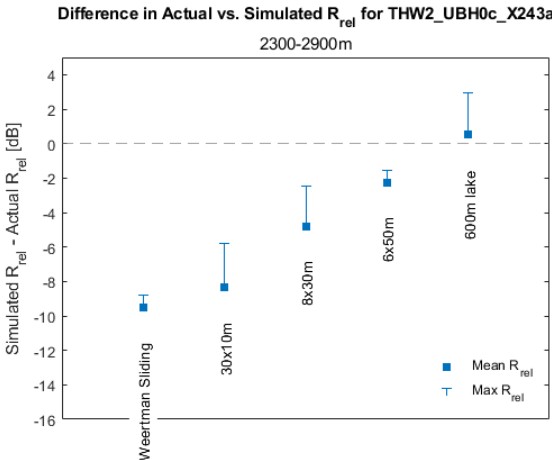

**Figure 11.** Mean difference between actual and simulated $R_{rel}$ over the area of interest at $2300 - 2900m$, for all simulations of THW2/ UBH0c/ X243a. Positive error bars show the difference between actual and simulated peak $R_{rel}$ over the same along-track range.

we have shown in our hypothetical simulations can reduce $R_{rel}$ by $\sim 6\,dB$. In future iterations of the simulator, we would enable heterogeneity in $\sigma_{bed}$ in order to capture this type of variation explicitly.

$R_{rel}$ near $2000m$ along track in all simulations was $\sim 10\,dB$ lower than actual reflectivity from THW2/ UBH0c/ X243a (Fig. 10). This is coincident with a very steep topographical feature in the THW2/ UBH0c/ X243a radargram. Steep slopes

such as this likely represent a limitation of our simulation approach. The surface representation using $5m$ flat facets will inherently direct more reflected energy away from the antenna position than a real surface. Therefore, caution is imperative when interpreting results near steep topography.

There are several additional limitations of our Stratton-Chu simulation method for testing sub-glacial hypotheses. Our choice of simulation radius $R = 300m$ explicitly limits the impact of range migration and clutter. A more complete simulation incor-

porating a larger $R$ would be highly beneficial to testing for sub-glacial water. Larger $R$ would allow simulations in thicker ice, with longer focusing apertures. Specularity content, which is often used to distinguish between distributed and channelized water features (Schroeder et al., 2015; Rutishauser et al., 2018; Schroeder et al., 2013), relies on focusing apertures up to $2km$. Future work should combine additional computing power with simulated facet roughness (Gerekos et al., 2023). This approach will provide additional realism and efficiency required to support simulated specularity content.

Our 3 dimensional model for topography was derived from a single along-track IPR flight line, which enables high confidence and sufficient resolution ($7.5m$) for ice geometry at nadir. However, this has the obvious limitation of requiring a previous flight line to build our simulation. We lack similar observation density across-track, leaving low-resolution ($500m$) open-source DEMs (such as BedMachine V2 (Morlighem, 2020)) as our best option for approximating off-nadir features. This asymmetry is not problematic for replicating basic topography at nadir, but the lack of realistic off-nadir features reduces the



sharpness of the focusing algorithm. We can see this effect by comparing the image quality in Fig. 8a to simulated results in Fig. 8b or Fig. 9a. This limitation also clearly reduces our ability to assess hypotheses involving any off-nadir targets.

Finally, our material model in this study was deliberately simple. In reality, there is a much broader range of possible bed materials, including hydrated tills and debris-laden ice, with real dielectric constants ranging from $3 - 36$ (Christianson et al., 2016). Allowing for additional material heterogeneity in the dielectric model could improve the robustness of the simulated results, but may also increase ambiguity as a range of non-unique solutions arise. We also assumed constant radar attenuation in the ice. This assumption is reasonable given the short physical distances of our radar simulations, but simulations over greater distances may require introducing variation in the imaginary component of $\epsilon_{ice}$ to account for heterogeneous attenuation loss.

## 4  Conclusions

In the exercise presented here, we optimized a radar simulation technique developed by Gerekos et al. (2018) to study the theoretical IPR response from sub-glacial systems. The simulator incorporates the Stratton-Chu integral and Linear Phase Approximation to efficiently estimate backscattered radar signal from simulated targets. Through a series of hypothetical simulations, we demonstrated the impact on relative reflectivity from rounded Röthlisberger channels or specular flat canals surrounded by bed materials of varying roughness. These simulations confirmed that reflectivity is highly dependent on both the size and cross-sectional shape of the sub-glacial water structure. Our results can be applied for inferring the presence, size, and structure of sub-glacial water bodies from IPR data in a more robust and sophisticated way than previous methods.

In our simulations of THW2/ UBH0c/ X243a, we demonstrated the simulator's utility in testing relevant hypotheses in sub-glacial hydrology. A large water structure could produce the elevated reflectivity beneath Thwaites Glacier, in a region coinciding with one channel route proposed by Hager et al. (2022). However, the radar signature is more consistent with a wide area of distributed water, such as a series of flat canals or a sub-glacial lake. Canals would average $> 30m$ in width and cover at least half the area for $600m$ in the along-track dimension. Although we do not see a Röthlisberger channel at this precise location, our findings do not preclude Röthlisberger channelization further upglacier. Further simulations of new and existing IPR survey data could help characterize the extent of upglacier channelization.

The method we outline has broad applicability for studying the basal environment of large glaciers. As we have shown, the simulation methodology can offer useful constraints when testing sub-glacial hypotheses. Scientific intuition, additional data inputs, and more computational power will improve the promise of this technique. Given the expense and challenging logistics of collecting IPR data, a forward model capable of predicting optimal locations for sub-glacial survey targets, instead of modeling existing flight lines, is an area of interest for future work. In such a forward model, computing resources must be optimized by strategically constraining parameter sets, and performing sensitivity tests for many of the variables considered here.

*Data availability.* All referenced data in this paper are made available at: https://doi.org/10.5281/zenodo.8165256





**Appendix A: Along-Track Focusing**

**A1    Along-Track Focusing**

We produce the focused radargram ($\xi_f$) by processing in along-track blocks. For a given fast-time range bin $\tau_j$, the block size, $L_a$, is chosen such that range migration for a target equals 3 fast-time samples, as depicted in A1. Thus it is important to note
that the block size increases with depth, and simulation radius $R$ must be greater than the maximum anticipated $L_a$.

To process a block centered at slow-time $a_0$, with depth $\tau_j$, we begin with a block of length $2L_a$ from the range compressed radargram ($\xi_{RC}$) as shown in A1. We calculated a 1-D reference function ($\phi$), representing the Doppler phase modulation as the antenna travels across the aperture in slow-time $a$ (A1) Peters et al. (2007); Legarsky et al. (2001); Hélière et al. (2007). The amplitude term, $b$, in A1 is used in real-world IPR processing to account for along-track variations in instrument gain, aircraft
motion, and to attenuate high Doppler frequency contributions at long apertures Legarsky et al. (2001); Peters et al. (2007). Our simulations do not contend with non-ideal flight or instrumentation variables, and we therefore use a simple Hamming window of width $L_a$ for suppression of higher frequency sidelobes.

$$\phi_j(a) = b(a)e^{-i2f_c\tau(a)}\Big|_{a=a_0-L_a/2}^{a=a_0+L_a/2} \tag{A1}$$

$$\xi_f(a,\tau) = \int\limits_{-L_a/2}^{L_a/2} \xi_{RC}(a+a',\tau)\,\Phi(a',\tau)^*da' \tag{A2}$$

The data block and reference function are Fourier transformed and convolved in the frequency domain (Fig. A2). The result is transformed back to the slow-time domain via inverse Fourier transform. Because the reference function $\phi$ is tuned at the center of the original block, the middle of the final block is better focused than the edges Hélière et al. (2007). For this reason, only half of the final block (length $L_a$) is written to the focused radargram $\xi_f$ (A1). This process is repeated for each along-track block, at all fast-time range bins until a complete focused radargram, $\xi_f$, is formed.



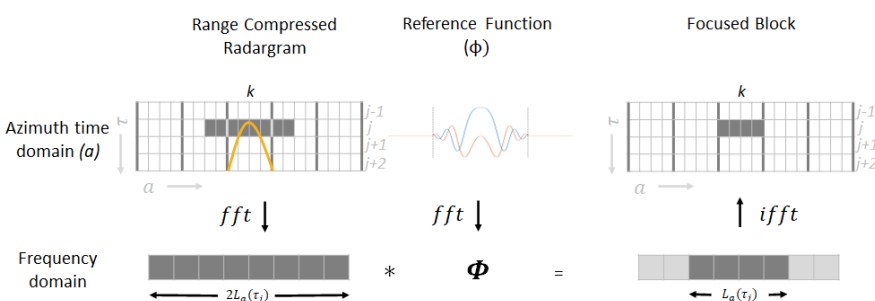

**Figure A1.** Schematic representation of along-track (azimuth) focusing for a single block at discrete fast-time increment $\tau_j$ and azimuth block $k$ in a simulated radargram. The orange hyperbola superimposed on the range compressed radargram illustrates the theoretical range migration of a target with shortest fast-time range $\tau_j$. We select an aperture length $L_a$ such that range migration spans 3 sample cells ($\tau_j$, $\tau_{j+1}$, and $\tau_{j+2}$).



## 455    A2    List of Variables

**Table A1.** List of variables used in this manuscript

| Symbol | Definition | Units |
|--------|------------|-------|
| $a$ | Time scale for radar observations along track (azimuth) | $s$ |
| $b$ | reference function amplitude scaling function | |
| $B_w$ | Radar bandwidth | $Hz$ |
| $c$ | Speed of light in free space | $m\,s^{-1}$ |
| $c_h$ | Channel height | $m$ |
| $c_w$ | Channel width | $m$ |
| $d_{ice}$ | Nominal ice thickness | $m$ |
| $\bar{E}_{b,n}$ | Backscattered electric field strength from $S_{ice}$ to radar antenna | $V\,m^{-1}$ |
| $\bar{E}_{i,n}$ | Incident electric field strength from radar antenna to $Sice$ | $V\,m^{-1}$ |
| $\bar{E}_{r,n}$ | Reflected electric field strength from $S_{bed}$ to $Sice$ | $V\,m^{-1}$ |
| $\bar{E}_{t,n}$ | Transmitted electric field strength from $S_{ice}$ to $Sbed$ | $V\,m^{-1}$ |
| $f_c$ | Radar central frequency | $Hz$ |
| $f_s$ | Sampling frequency | $Hz$ |
| $g$ | Radar chirp signal | |
| $h$ | Aircraft height | $m$ |
| $j$ | Fast-time ($\tau$) incremental index | |
| $k$ | Azimuth time ($a$) incremental index | |
| $\bar{k}_{i,n}$ | Wavevector from radar antenna to facet $n$ on $Sice$ | |
| $\bar{k}_{r,n}$ | Reflected wavevector from facet on $S_{bed}$ to $S_{ice}$ | |
| $\bar{k}_{t,n}$ | Transmitted wavevector from facet $n$ on $S_{ice}$ to $S_{bed}$ | |
| $L_a$ | Aperture length for SAR focusing | $m$ |
| $l_c$ | Roughness correlation length | $m$ |
| $l_f$ | Facet length | $m$ |
| $l_{f2}$ | Secondary facet length for Röthlisberger channels | $m$ |
| $PRF$ | Pulse repetition frequency | $Hz$ |
| $R$ | Simulation radius | $m$ |
| $R_{abs}$ | Absolute reflectivity | $dB$ |
| $R_{abs,0}$ | Absolute reflectivity of frozen bed | $dB$ |
| $R_{pl}$ | Pulse limited radius | $m$ |
| $R_{rel}$ | Relative reflection coefficient | $dB$ |



**Table A2.** List of variables, continued

| Symbol | Definition | Units |
|---|---|---|
| $r_{air}$ | one-way travel distance through air from radar to ice surface | $m$ |
| $r_{ice}$ | one-way travel distance through ice from ice surface to target | $m$ |
| $RCM$ | range migration, number of discrete fast-time cells | |
| $S_{bed}$ | Discretized bed surface | |
| $S_{ice}$ | Discretized ice surface | |
| $S^{BM}$ | 2-D topography matrix derived from BedMachine V2 (ice or bed) | $m$ |
| $S^{rad}$ | 2-D topography matrix derived directly from radar data | $m$ |
| $S^{rough}$ | 2-D matrix representing random isotropic Gaussian roughness | $m$ |
| $T_r$ | Radar receiving window | $s$ |
| $T_s$ | Radar pulse length | $s$ |
| $v$ | Aircraft velocity | $m\,s^{-1}$ |
| $w$ | Across track quadratic weighting function (values [0,1]) | |
| $x$ | Cartesian spatial coordinate, across track | $m$ |
| $y$ | Cartesian spatial coordinate, along track | $m$ |
| $z$ | Cartesian spatial coordinate, elevation | $m$ |
| $z_0$ | Polynomial interpolation of ice elevation, approximating aircraft drape | $m$ |
| $\epsilon_{H_2O}$ | Relative dielectric constant of water | |
| $\epsilon_{ice}$ | Relative dielectric constant of ice | |
| $\epsilon_{rock}$ | Relative dielectric constant of rock | |
| $\eta_{H_2O}$ | Refractive index of water | |
| $\eta_{ice}$ | Refractive index of ice | |
| $\eta_{rock}$ | Refractive index of rock | |
| $\theta$ | Channel orientation angle relative to $x$-axis | $rad$ |
| $\lambda$ | Radar center wavelength | $m$ |
| $\xi_f$ | Focused radargram | $V\,m^{-1}$ |
| $\xi_{raw}$ | Raw radargram | $V\,m^{-1}$ |
| $\xi_{RC}$ | Range compressed radargram | $V\,m^{-1}$ |
| $\sigma$ | Roughness amplitude for surface $i$ | $m$ |
| $\tau$ | Fast-time scale for returned radar echoes | $s$ |
| $\tau_s$ | Fast-time value for radargram ice surface reflection | $s$ |
| $\phi$ | 1-D reference function for radargram focusing | |
| $\Phi$ | Slow-time Fourier transform of reference function $\phi$ | |



*Author contributions.* CP prepared the original draft, wrote modifications to the simulator, led the design and execution of experiments, and participated in the field work to collect IPR data from Thwaites Glacier. CG contributed the original simulator software, assisted in experimental design, model validation, and provided significant editorial support. MS provided oversight and partial funding, as well as significant editorial contributions. LB provided important context for the project conceptualization, made significant editorial contributions,
and led the field team responsible for IPR data collection over Thwaites Glacier. DB provided leadership support, funding, and programmatic resources for the investigation and IPR data collection over Thwaites. WSL provided funding and logistics support for the Thwaites IPR data collection, as well as editorial contributions. EA provided significant editorial review and advisory support. CKL made a significant contribution to planning and execution of the Thwaites Glacier IPR survey, provided input on the simulation experimental design, and provided editoral support. JS participated in the collection and validation of IPR data from Thwaites Glacier.

*Competing interests.* The authors declare they have no conflicts of interest.

*Acknowledgements.* This research was supported by the National Aeronautics and Space Administration (Award: 80NSSC20K1134), Korea Institute of Marine Science & Technology Promotion (KIMST) funded by the Ministry of Oceans and Fisheries (RS-2023-00256677; PM23020), and the Vetelsen Foundation. KIMST and Canadian Helicopters Limited (CHL) provided additional logistics, equipment, and personnel supporting IPR data collection over Thwaites Glacier. Finally, we would like to acknowledge individual contributions from Dillon
Buhl, Greg Nguyen, Dr. Kirk Scanlan. Their expertise was invaluable to completing this manuscript.





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
