# Peer review of "Characterizing Sub-Glacial Hydrology Using Radar Simulations"

_EGUsphere, 2023_

## Referee Comment (RC2)

The manuscript offers a compelling blend of electromagnetic modelling and radar data analysis in the context of Thwaites Glacier's subglacial drainage pathway. While the results hold promise for enhancing our knowledge of glacier subglacial hydrology, however it needs to be improved considering the suggestions below.

**Some of the Major issues to be addressed, they are.**

- If the manuscript aims to identify the subglacial water arrangement by eliminating alternative hypotheses, it is imperative that the comprehensive permittivity values for both water and the glacier bed cover the entire spectrum found in existing literature, thus reproducing the full range of reflectivity associated with these parameters.
- The paper primarily focused on the regional scale of Thwaites' downstream water system. However, it is imperative to acknowledge the broader applicability of the model for a more extensive catchment when considering its practical utility.

**Minor comments:**

- Line 25: Better to add more recent citations.
- Line 32: Missing citations
- In the introduction section where the objectives are outlined, it is important to include the potential applications of the model to other catchments, thereby emphasising its versatility beyond the specific case of Thwaites Glacier.
- Providing a dedicated "Study Area" section, rather than including it in the introduction, would offer a more comprehensive understanding of the research area.
- Additionally, separating the "Results" and "Discussion" sections would enhance the clarity and structure of the paper. It would improve the presentation if all figures and tables were enclosed by borders for a more organized and visually clear layout.
- In the conclusion section, it would be beneficial to provide an overview of future work, offering the potential research directions and developments to be pursued.

---

## Author Response (AR1)

We appreciate both reviewers' insightful comments on our manuscript "A Simulation Approach to Characterizing Sub-Glacial Hydrology". These will were helpful in improving the final product.

The reviewers' original comments are listed below. A summary of revisions made to the manuscript addressing each comment are listed in *red*:

Reviewer 1 - Major issues:

- The range of material properties for the bed and water used in the study seem too narrow given the literature cited in the paper. For example, Peters 2005 included reflectivity differences between water (including groundwater) and frozen bedrock that differ by 26 dB without invoking any change in bed roughness or geometry. If you look at Christiansen 2016 and Tulaczyk and Foley 2020 (https://doi.org/10.5194/tc-14-4495-2020) these values are also in the 25 - 27 dB range. If the manuscript seeks to diagnose the subglacial water configuration by excluding other hypotheses, then the complex permittivity for both water and the bed explored should span the full range of this literature (and reproduce that range of reflectivities).
    - *In the Discussion, we have added a figure and text addressing the range of possible dielectric constants that were not explored in this study. We provide justification as to why much of this range is unlikely due to the observed reflectivity profile and physical environment. We have also adjusted the language in our Results, Discussion, and Conclusions to indicate that we have eliminated a Röthlisberger channel as the possible source of the bright echo on THW2/UBH0c/X243a. We state that the echo is consistent with canals or a sub-glacial lake, while acknowledging that we have not explored all possible alternatives required to make a conclusive diagnosis.*

- Similarly, the range of bed roughness considered in the study is also small compared to the literature cited by the paper. Again, Peter 2005 shows roughness-based reduction in reflectivities that are as high as 20 dB. For the reasons described above, I'd expect this paper to explore roughesses losses at that scale as well.
    - *In the Discussion section, we detail the limitations of our roughness evaluation in this study. We specifically address the Peters 2005 paper, which calculated theoretical losses due to roughness up to 20dB over very short correlation lengths, but measured roughness over distances exceeding 1km. As the reviewer suggests, more work should be completed to evaluate the impact of roughness on radar returns, which we suggest as a direction for future work. As with the previous point, we acknowledge in the Discussion and Conclusions that we have not explored all possible combinations of material and geometric parameters.*

- The authors present a simulation that is focusable (and focused) which should allow them to probe the specularity of bed echoes for each of the hypothesized configurations. Since, as the paper mentions, this was the key observable in classifying the downstream water system of Thwaites as concentrated rather than distributed, it would seem incumbent on the authors to address whether their interpretations of the (inherently more ambiguous) reflectivity signal would also explain that larger catchment-wide specularity signature.
  - *We have addressed this limitation of our work in the Discussion section. While we concur that simulations replicating specularity content would be helpful, the aperture lengths required (~2km) exceed the capacity of our computational resources.*

- Line 180: The authors state that they "confine" themselves to Röthlisberger channels and flat canals for this study. That is a fine choice to support the claim (if it survives addressing the issues raised above) that the bright spot (and downstream water network) is likely not a canonical Röthlisberger channel. However, in order to claim (as the authors do in their abstract and conclusion) that the wanted body is "distributed" (which has a specific subglacial hydrological meaning and implication for modeling) they would need to also address other "concentrated" water geometries and exclude them as well. These include Nye Channels, Creytes & Schoof water sheets (https://doi.org/10.1029/2008JF001215) an other concentrated/efficient water configurations (https://doi.org/10.1098/rspa.2014.0907). Otherwise the authors should limit themselves to falsifying the hypothesis that the bright echo on THW2/UBH0c/X243a is a canonical Röthlisberger channel and remove language like "We ultimately conclude the bright reflector from our IPR flight line is more likely a broad area of wide distributed water, such as a series of flat canals or sub-glacial lake" which cannot be supported by a study that does not consider other "concentrated " water geometries.
  - *As the reviewer suggested, we have acknowledged in both the Methodology and Discussion sections that other water geometries could exist, which were outside the scope of our study. Our language now indicates that the reflector cannot be a Röthlisberger channel, but is consistent with a distributed geometry such as a series of flat canals or a lake. We leave the possibility open that the reflectivity profile could match other unexplored configurations.*

Reviewer 2 – Major issues:

- If the manuscript aims to identify the subglacial water arrangement by eliminating alternative hypotheses, it is imperative that the comprehensive permittivity values for both water and the glacier bed cover the entire spectrum found in existing literature, thus reproducing the full range of reflectivity associated with these parameters.
  - *This comment is similar to the first issue from Reviewer 1*

- The paper primarily focused on the regional scale of Thwaites' downstream water system. However, it is imperative to acknowledge the broader applicability of the model for a more extensive catchment when considering its practical utility.
  - *We have included language in the Abstract, Introduction, and Conclusions indicating the broader applicability of the simulation technique. We also re-arranged several sections of the paper at the reviewer's suggestion. The Introduction now focuses on the technique itself, while the Thwaites study area is presented much later as an example use case.*

Reviewer 1 - Minor Issues:

- Abstract Line 1: Depending on its configuration water can either enhance or reduce sliding and/or retreat.
  - *We have adjusted the language as suggested by the author.*

- Abstract Line 3: Given the recent paper by Schlegel et al. ( https://doi.org/10.1017/aog.2023.2) you may want to consider the use of IPR here.
  - *The nomenclature throughout the paper was changed from "IPR" to "RES" or "Radar Echo Sounding", to be consistent with Schlegel et al.*

- Table 1: 1.71 MW seems like an extremely high value.  It's unusual to report a post-processing number for transmit power and presenting it in this way could confuse readers significantly.
  - *We have adjusted the table to reflect both MARFA native and simulated power. A footnote was added to explain the difference.*

Reviewer 2 - Minor issues:

- Line 25: Better to add more recent citations.
  - *We have added more recent citations for the relationship between hydrology, shear stress, and ice rheology as suggested (Line 28).*

- Line 32: Missing citations
  - *We added several citations establishing IPR as a technique in the study of hydrology (Line 35).*

- In the introduction section where the objectives are outlined, it is important to include the potential applications of the model to other catchments, thereby emphasising its versatility beyond the specific case of Thwaites Glacier.
  - *See our response to the second bullet under "Major Issues". We believe the revised manuscript emphasizes the utility beyond Thwaites and the introduces the Thwaites flight line only as an example use case.*

- Providing a dedicated "Study Area" section, rather than including it in the introduction, would offer a more comprehensive understanding of the research area.
  - *At the reviewer's request, a dedicated Study Area section (4.1) was included when introducing the Thwaites Glacier example use case.*

- Additionally, separating the "Results" and "Discussion" sections would enhance the clarity and structure of the paper. It would improve the presentation if all figures and tables were enclosed by borders for a more organized and visually clear layout.
  - *We have re-organized several sections of the paper to be consistent with this comment. The paper now has the following structure:*
    - *Introduction*
    - *Simulation Methodology*
      - *Dielectric Material Model*
      - *Geometric Model*
      - *Surface Roughness*
      - *Simulation Radius*
      - *Channel Geometry*
      - *Simulated Data Processing*
      - *Hypothetical Simulation Cases*
    - *Results - Hypothetical Simulation Cases*
    - *Application to Thwaites Glacier*
      - *Study Area*
      - *Method*
      - *Results*
    - *Discussion*
    - *Conclusions*

- In the conclusion section, it would be beneficial to provide an overview of future work, offering the potential research directions and developments to be pursued.
  - *We added language in both the Discussion and Conclusions sections indicating possible directions for future work.*

---

## Author Response (AR2)

We appreciate the reviewer's feedback regarding our conclusions on the sub-glacial water structure beneath Thwaites. The reviewer accurately points out that we have not eliminated all possible water geometries, and therefore it is inappropriate to imply definitive diagnosis of the water body in THW2/UBH0c/X243a as distributed without further evidence. We agreed with the reviewer's original feedback on this point, and apologize if our original revisions were not sufficiently clear.

To address this, we have further revised the abstract, discussion, and conclusions to state that "of the scenarios we tested", the reflector in THW2/UBH0c/X243a appears "most consistent with" a series of canals or a lake.  We also explicitly state that our simulations were not exhaustive of all possible subglacial water configurations. We have also removed language referring to the water bodies in our simulations as "distributed".

We agree that the conclusions drawn from our simulations should be specific. However, we disagree with the reviewer's insistence that we remove all reference to lakes or canals, and that our work does not demonstrate consistency with these water structures. Our simulations clearly demonstrate that scenarios involving canals and/or lakes can replicate the magnitude of the reflectivity signature. Therefore, while we have agreed to carefully qualify our concluding language, we consider it is appropriate to highlight the consistency between the radar reflectivity and our simulations involving canals or lakes.

---

## Author Response (AR3)

Once again, we thank the reviewer for clarifying their feedback on the hypothesized structure for the sub-glacial reflection beneath Thwaites Glacier. We hope the following adjustments to our manuscript will satisfy any remaining concerns the reviewer may have regarding the manuscript:

- We have added a new figure, (Figure 12) that displays the observed specularity content data from THW2/ UBH0c/ X243a. We note the manuscript's primary focus was on the modeling methodology. In previous drafts specularity content was only briefly discussed because we could not simulate it well. However, the reviewer is correct that it does offer additional evidence for the nature of the reflector in question. The observed specularity content in the area of interest on THW2/ UBH0c/ X243a is elevated, which may further support the hypothesis of a lake or canals.
- We added an additional example of distributed sub-glacial structures to Figure 6 that are within the downstream zone on Thwaites as proposed by Schroeder et al. (2013). It is not our intent to challenge the hypothesis that Thwaites' downstream hydrology is dominated by channelized structures, however, there is at least one example of a proposed sub-glacial lake within this region (Smith et al. 2017).

We would also like to clarify that the downstream hydrological system proposed by Schroeder et al. (2013) was based on RES data collected in 2003-2004 covering the entire Thwaites catchment. Schroeder et al. (2013) followed several flow lines from the upper region of Thwaites to the grounding zone, and hypothesized that "a significant decrease in specularity accompanying a comparatively high relative echo strength" was evidence for a hydrological transition from primarily distributed canals to Röthlisberger channels.

Our RES survey is comprised of new data collected in 2022 over a relatively confined geographic footprint in the downstream Thwaites region. We do not feel the broader hypothesis from Schroeder et al. (2013) is compromised by isolated reflectors with high specularity and reflectivity, such as the reflection we have identified as an example from THW2/ UBH0c/ X243a. However, as the reviewer notes, we demonstrated through our simulations that Röthlisberger channels are unlikely to produce reflectivity comparable to canals.

Further, our manuscript simulated the sub-glacial environment for a 4km segment from a single flight line. Our simulations and the RES data (both specularity and reflectivity) could be consistent with canals or lakes, as we have shown. We consider it inappropriate to extrapolate our conclusions to the wider Thwaites hydrological system, although of course a future objective will be to expand our simulation efforts to a wider geographic footprint. We hope the reviewer agrees with this approach.